# Be$^+$ assisted, simultaneous confinement of more than 15000 antihydrogen atoms

R. Akbari[1], L. O. de Araujo Azevedo[2], C. J. Baker [3], W. Bertsche[4,5], N. M. Bhatt [3], G. Bonomi [6], A. Capra [7], I. Carli [7], C. L. Cesar [2], M. Charlton [3], A. Cridland Mathad [3], A. Del Vincio[6], D. Duque Quiceno [1,7], S. J. Eriksson[3], A. Evans[1,7], J. Ewins[1], J. Fajans [8], T. Friesen[9], M. C. Fujiwara[7], L. M. Golino [3], M. B. Gomes Gonçalves[3], J. S. Hangst [10] ✉, M. E. Hayden[11], D. Hodgkinson [8], C. A. Isaac[3], A. J. U. Jimenez [9], S. A. Jones [12], S. Jonsell [13], N. Madsen [3] ✉, V. R. Marshall[10], J. T. K. McKenna[4], T. Momose [1,7,14], J. Nauta [3,15], A. N. Oliveira [4], J. Peszka [3], A. Powell [9], C. Ø. Rasmussen [15], T. Robertson-Brown [3], F. Robicheaux [16], R. L. Sacramento[2], E. Sarid[17,18], J. Schoonwater [3], D. M. Silveira[2], J. Singh [4], G. Smith [1,7], C. So[7], S. Stracka [19], J. Suh [9], A. G. Swadling[9], T. D. Tharp[20], K. A. Thompson[3], R. I. Thompson [7,9], E. Thorpe-Woods [3], M. Urioni [6], D. P. van der Werf [3], P. Woosaree [9] & J. S. Wurtele [8]

Antihydrogen, the bound state of a positron and an antiproton, is the only pure anti-atomic system ever studied. It is produced exclusively in the laboratory, as it has never been observed in nature. This unique system is of great interest for searching for tentative differences between matter and antimatter. Antihydrogen has been routinely trapped since 2010 and accumulated since 2017, enabling, for example, the first precision spectroscopic study of the anti-atom in 2018 and the first observation of the influence of gravity in 2023. Here we report an eight-fold increase in the trapping rate of antihydrogen, enabled by sympathetic cooling of positrons with laser-cooled beryllium ions. With beryllium sympathetic cooling, we now accumulate over 15000 antihydrogen atoms in under seven hours. This technique transforms our ability to study systematic and sidereal effects in existing experiments while paving the way for studies that would otherwise remain out of reach.

Exotic atoms, or atomic systems that do not naturally occur (e.g., positronium[1], muonic atoms[2], and antiprotonic helium[3], are powerful tools to study fundamental physics. Two major challenges in working with such species are the rate at which they can be synthesised and their typically short life- or confinement times. Antihydrogen was initially synthesised in 1995[4], and low energy antihydrogen, useful for experimentation, debuted in 2002[5]. Antihydrogen atoms were first trapped in 2010[6], and accumulation (where antihydrogen is repeatedly added to an already trapped sample) has been in use since 2017[7]. The continuous improvement in the number of trapped antihydrogen

atoms enabled a number of measurements on antihydrogen, among these the $2 \times 10^{-12}$ precision measurement of the frequency of the 1S-2S transition[8], the demonstration of laser cooling using the Lyman-alpha (1S-2P) transition[9], and the first measurement of the influence of gravity on antihydrogen[10]. Some goals that remain are to attain the seminal precision of $4 \times 10^{-15}$ achieved in hydrogen 1S-2S spectroscopy[11], to improve the precision of our measurements of the ground state hyperfine splitting[12] and the gravitational acceleration[10], and to achieve the first measurements of the antihydrogen Rydberg constant, Lamb-shift, and antiproton charge radius. An increase in the

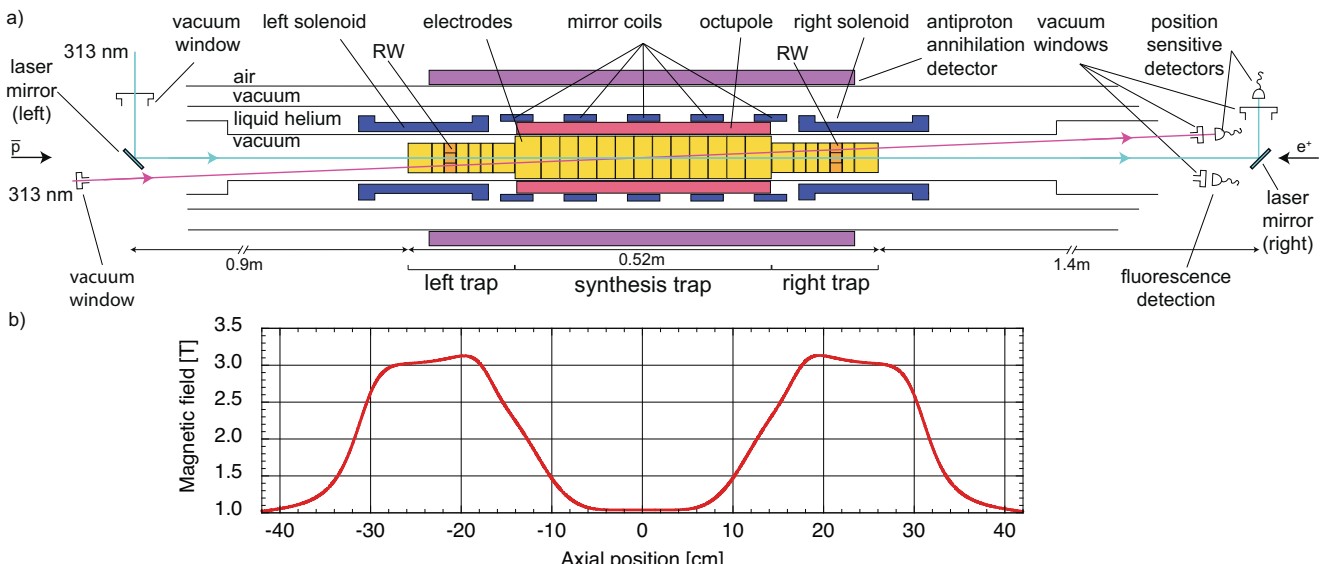

**Fig. 1 | The experimental apparatus and the axial magnetic field profile. a** Cross section of the central part of the ALPHA-2 experimental apparatus. In yellow and orange are shown the electrodes used to generate the electric fields for the Penning-Malmberg trap. The superconducting magnets that generate the magnetic-minimum trap (octupole and mirror coils) and boost the axial magnetic field to 3 T in the two side regions (left and right solenoids) are shown in blue and pink. The external, 1 T solenoid is not shown. Lasers for Be+ Doppler cooling can be injected along an off-axis path at 2.3 degrees through vacuum windows at either end of the apparatus (purple path), or along an on-axis path through a vacuum window onto movable 45-degree mirrors that can be positioned at either end of the

central apparatus (cyan path). The orange electrodes in the left and right regions are azimuthally segmented to allow for the rotating wall (RW) technique (see text). The Be+ source (not shown) is placed on the same linear translator as the left side mirror[20]. Microchannel plate/phosphor assemblies used for imaging extracted particles (not shown) are placed on the same translators as the mirrors on both sides. The central portion of the drawing is to scale except for the annihilation detector, whose axial length is marked by purple bars. Position sensitive detectors are used for stabilizing the cooling laser beams. **b** The corresponding longitudinal magnetic field (on-axis) that is maintained during Be+ assisted H̄ accumulation.

trapped antihydrogen accumulation rate facilitates higher accuracy measurements and the implementation of enhanced experimental procedures. Experimental and theoretical evidence suggests that for the synthesis technique we use, in which cold plasmas of positrons and antiprotons are merged in a Penning-Malmberg trap, the antiprotons reach thermal equilibrium with the positron plasma before combining to form antihydrogen[7,13]. Achieving the lowest possible temperatures of the positron plasmas is therefore key to efficiently synthesising cold, trappable antihydrogen. The pioneering technique presented below was motivated by and confirms this premise.

Here we report on the development, implementation, and successful use of a new technique using laser-cooled beryllium ions (Be+) to sympathetically cool positrons (e+) during the 2023 and 2024 antiproton (p̄) experimental periods at CERN. This technique has led to a near eight-fold increase in antihydrogen (H̄) accumulation rates. Additionally, the first systematic studies with this technique have offered new, quantitative insights into the H̄ synthesis process and its dependence on the e+ plasma temperature. We demonstrate through direct control the key importance of the e+ temperature when attempting to synthesise H̄.

Antihydrogen is synthesised by merging cold plasmas of p̄ and e+ in a cylindrical Penning-Malmberg trap, where strong (1–3 T) axial magnetic fields confine the charged particles transversely, and co-axial electrodes can be individually biased to confine the particles axially (Fig. 1). Antiprotons are sourced from the CERN Antiproton Decelerator (AD), whose latest addition is the ELENA (Extra Low ENergy Antiproton) decelerator that, since 2021, delivers about $10^7$ p̄ at an energy of 100 keV every two minutes[14]. Preparation of the p̄ and e+ plasmas for H̄ synthesis has been described elsewhere[7]. After the final steps of this preparation have taken place in the left trap (Fig. 1), about $10^5$ p̄ in an ellipsoidal cloud of radius ~0.4 mm with a temperature of ~100 K are merged in 2–3 s with a cloud of about 3 million e+ to synthesise H̄ in the synthesis trap region (Fig. 1). The coldest anti-atoms

are subsequently trapped by a Ioffe-Pritchard type, magnetic-minimum trap with a trapping depth of about 0.5 K (we typically use K as an energy unit). Prior to the work presented here, the e+ first cooled via the emission of cyclotron radiation in the strong longitudinal magnetic field, followed by evaporative cooling[15], adiabatic expansion cooling[16], or both. Typical densities and temperatures were ~$6.5 \times 10^7$ cm$^{-3}$ and ~18 K, and the reported trapping from these experiments was $14.4 \pm 0.8$ trapped anti-atoms per synthesis cycle[7]. We typically capture every second AD/ELENA ejection, which results in a synthesis cycle that has a duration of about four minutes. For comparison, in 2010, on average, anti-atoms were trapped at a rate of 0.1 per 15 min-long synthesis cycle[6]. This initial performance was improved by lowering the radiation temperature in the trap (e.g., by adding radiation shielding and electronic noise shielding) and adding e+ evaporative[15] and adiabatic cooling[16] steps. Additional tuning led to the hitherto highest trapping rate of ~20 trapped anti-atoms per synthesis cycle[8] with a four-minute repetition. Many incremental improvements were correlated with lower e+ temperatures[7]. This correlation between e+ temperature and trapping rate, and the original demonstration of Be+ cooled e+ by Jelenković et al.[17] motivated a theoretical and computational study of the feasibility of using a laser-coolable ion species to sympathetically cool the e+ even further[18].

Motivated by the promising outcome of the theoretical study, we implemented a system to load Be+ ions directly into our H̄ synthesis trap using pulsed laser ablation[19]. More recently, we showed that a cloud of about $10^5$ laser-cooled Be+ ions can cool millions of e+ to temperatures of around 7 K, or about a factor of 2.5 below the temperature observed without the laser-cooled ions[20]. The achievable temperatures are well above the Be+ Doppler cooling limit (~0.5 mK) due to the inefficiency of Coulomb coupling between light and heavy species and centrifugal separation between the different mass species that can completely decouple the species at low enough temperatures[18,21],

To make the sympathetic cooling procedure compatible with $\bar{\text{H}}$ synthesis required modifications of our protocols that shortened the preparation time of the sympathetically cooled $e^+$ plasmas to 100 s (in our earlier work[20], we used up to four minutes to prepare the cryogenic $e^+$) and increased the shot-to-shot reproducibility of the $Be^+$ plasma preparations. These improvements were achieved by using a higher power 313 nm cooling laser and by adding an on-axis laser path to the experimental setup (Fig. 1), which originally contained only the off-axis laser path. The off-axis laser passes through the trap at a 2.3-degree angle with respect to the central axis[20]. The on-axis laser enters the trap using 45-degree mirrors mounted on transverse, linear translators placed at both ends of the trap. The translators allow for the mirrors to be moved away from the trap axis to permit particle injection into the trap. The $Be^+$ laser ablation source is located on the same linear translator as the left laser mirror (Fig. 1).

A key impediment to efficient sympathetic cooling is the fluctuation (up to a factor 10 for consecutive shots) in the number of $Be^+$ ions produced from laser ablation. A technique for controlling the density and size of $e^+$ and electron ($e^-$) plasmas was previously developed at ALPHA[22]. This technique, known as SDR-EVC (Strong-Drive Regime[23]−EVaporative Cooling), consists of applying a transverse, rotating electric field (also known as a rotating wall) to a plasma using azimuthally segmented electrodes, while simultaneously performing evaporative cooling. This results in a reproducible final plasma, independent of the parameters of the initial plasma, but requires an additional cooling mechanism to be effective. The new on-axis laser path enables laser cooling of $Be^+$ in the left and right traps (Fig. 1), which both contain the azimuthally segmented electrodes required for application of the SDR-EVC technique. Using this technique, we stabilise the number of $Be^+$ ions to a shot-to-shot variability of less than 14%. This is critical, as the efficacy of the sympathetic cooling of $e^+$ has been shown to be dependent on the relative number of $Be^+$ to $e^{+\,20}$. The SDR-EVC technique significantly simplified our $Be^+$ preparation procedure compared to[20]. This in combination with the shorter cooling times achieved with the higher cooling laser power were the key steps to achieving compatibility with antihydrogen synthesis.

Sympathetically cooled $e^+$ are prepared as follows: first, a pulse of laser ablated $Be^+$ ions is loaded into the left section of the trap (Fig. 1: 'left trap'). The ions are then laser cooled with -160 mW of 313 nm laser light, detuned −4 GHz from resonance with the $2s^2S_{1/2}$-$2p^2P_{3/2}$ cooling transition and directed along the on-axis, trap-coaxial path. Laser cooling continues while a strong-drive-regime (SDR) rotating wall is applied to the $Be^+$ plasma, and the potential well depth is lowered to evaporatively cool (EVC) the ions (Methods). These steps result in a - 500 K $Be^+$ plasma with -130k ions. Next, the $Be^+$ plasma is moved towards the axial centre of the synthesis trap (Fig. 1) to make space for the final $\bar{\text{p}}$ preparation[7]. While in the centre of the trap, the $Be^+$ plasma is laser cooled with the off-axis laser at −800 MHz detuning. Positrons are then ballistically transferred from the $e^+$ accumulator (see e.g., Ref. 24) through a magnetic beamline[25] and captured in the right section of the trap (Fig. 1: 'right trap'). The number of $e^+$ is subsequently stabilised using the SDR-EVC technique. The $e^+$ and $Be^+$ plasmas are now merged in the right trap where the axial field is -3 T to form a multi-species $Be^+/e^+$ plasma. No laser cooling is applied during this amalgamation. The merging of the plasmas heats up both species, but the $e^+$ cool through cyclotron radiation and sympathetically cool the $Be^+$. Subsequently, in a final step, discussed below, the $Be^+/e^+$ mixture is moved to the centre of the synthesis trap region, where $Be^+$ laser cooling is reapplied at a fixed, small detuning.

For $Be^+$ assisted synthesis of $\bar{\text{H}}$, the left and right solenoids and the magnetic-minimum trap are first energised and remain so throughout (cf. Fig. 1). The solenoids increase the magnetic field in the left and right trap regions to about 3 T to improve the lepton cyclotron cooling rate. Antiprotons are prepared as before, now in parallel with the $Be^+/e^+$ preparation, and are moved adjacent to the laser-cooled $Be^+/e^+$ mixture during the final steps of the sympathetic cooling procedure described above. A cloud of 150k $\bar{\text{p}}$ of radius 0.7 mm and a temperature of 150 K was typical in 2023.

For synthesis, the laser cooling is applied at a fixed small detuning while the $Be^+/e^+$ plasma is slowly mixed with the $\bar{\text{p}}$ cloud over 2-3 s using our potential merge technique[7]. During this merge, more than 95% of the $\bar{\text{p}}$ form $\bar{\text{H}}$ atoms, some of which are trapped[26]. The final laser cooling step is performed with the laser directed along either the on-axis path or the off-axis path. Using the off-axis path allows for diagnosing the $e^+$ plasma parameters by ejecting the $e^+$ to microchannel plate/phosphor assemblies mounted on the linear translators at both ends of the trap. Following $\bar{\text{H}}$ synthesis, all remaining charged particles are ejected from the trap using pulsed electric fields, before the $\bar{\text{H}}$ is released and counted. The number of $\bar{\text{H}}$ atoms is determined by detecting their annihilation on the internal walls. The loss and subsequent annihilation of trapped $\bar{\text{H}}$ can be effected in various ways. The simplest approach is to ramp down the octupole magnet of the magnetic-minimum trap. The annihilations are detected with our spatially resolving silicon-vertex detector[27] (SVD), which has a 99.5% efficiency for detecting annihilations. The main detector background is due to cosmic rays, but this background can be reduced by applying machine-learning (ML) methods[28] (Methods). For the analysis here, we use the detector triggers during synthesis as a measure of the total number of $\bar{\text{H}}$ atoms produced (the background is negligible during the 2−3 s synthesis). Employing the ML-based background suppression, we use reconstructed annihilation vertices to count $\bar{\text{H}}$ that has been trapped and subsequently released. The background rate of false positives of reconstructed annihilations (cosmic rays that are misinterpreted as $\bar{\text{p}}$ annihilations) is $0.0265 \pm 0.0002\,\text{s}^{-1}$, and the efficiency of fully reconstructed $\bar{\text{p}}$ annihilations is $72.1 \pm 0.1\%$. The background count and signal-to-background ratio thus depend on the time taken to release the trapped $\bar{\text{H}}$.

The various $Be^+$ and $e^+$ preparation steps were initially optimised, without using $\bar{\text{p}}$, by targeting the same $e^+$ density (-$10^8\,\text{cm}^{-3}$) and number ($3.4 \times 10^6$) that are used in our standard synthesis process[7] without $Be^+$. Here we followed our previous work[20], where we tuned the number of $Be^+$ ions while also changing the frequency detuning of the final laser cooling step to obtain the lowest $e^+$ temperatures. Due to the limited solid angle for fluorescence[20] detection, all of our plasma temperatures are determined using the parallel energy analyser technique[29,30]. The absolute temperature from this technique suffers from some unidentified systematic uncertainties[20] of up to 50% that do not influence the validity of relative changes when observing similar plasmas ejected from similar potential wells. The technique is unable to measure temperatures below -5 K[31].

## Results

Figure 2 shows the impact of changing the frequency detuning during the final laser cooling step (using the off-axis laser) on the $e^+$ density, $e^+$ temperature, number of $\bar{\text{H}}$ atoms synthesised, and number of $\bar{\text{H}}$ atoms trapped per synthesis cycle. All parameters are observed to depend on the detuning during the final laser cooling step. The densities observed are lower than our target stated above, which is partly due to radial expansion of the $Be^+/e^+$ mixture that may be due to torque from the off-axis cooling laser (see e.g., Ref. 32 for examples of this phenomenon) or due to expansion of the $Be^+/e^+$ mixture both before and during the $\bar{\text{H}}$ synthesis (the potential manipulations during synthesis cause expansion). It is a topic for further study to understand all the causes of plasma expansion observed at each step. In the end, we used the on-axis laser for our final cooling step for the majority of our 2023 $\bar{\text{p}}$ experimental time but reverted to the off-axis laser for the 2024 experimental time, as this allowed for better diagnostics of the final preparation. In Fig. 2 we observe a clear correlation between lower $e^+$ temperatures and an increase in the amount of $\bar{\text{H}}$ synthesised and trapped. This is in agreement with the expectation that the $e^+$

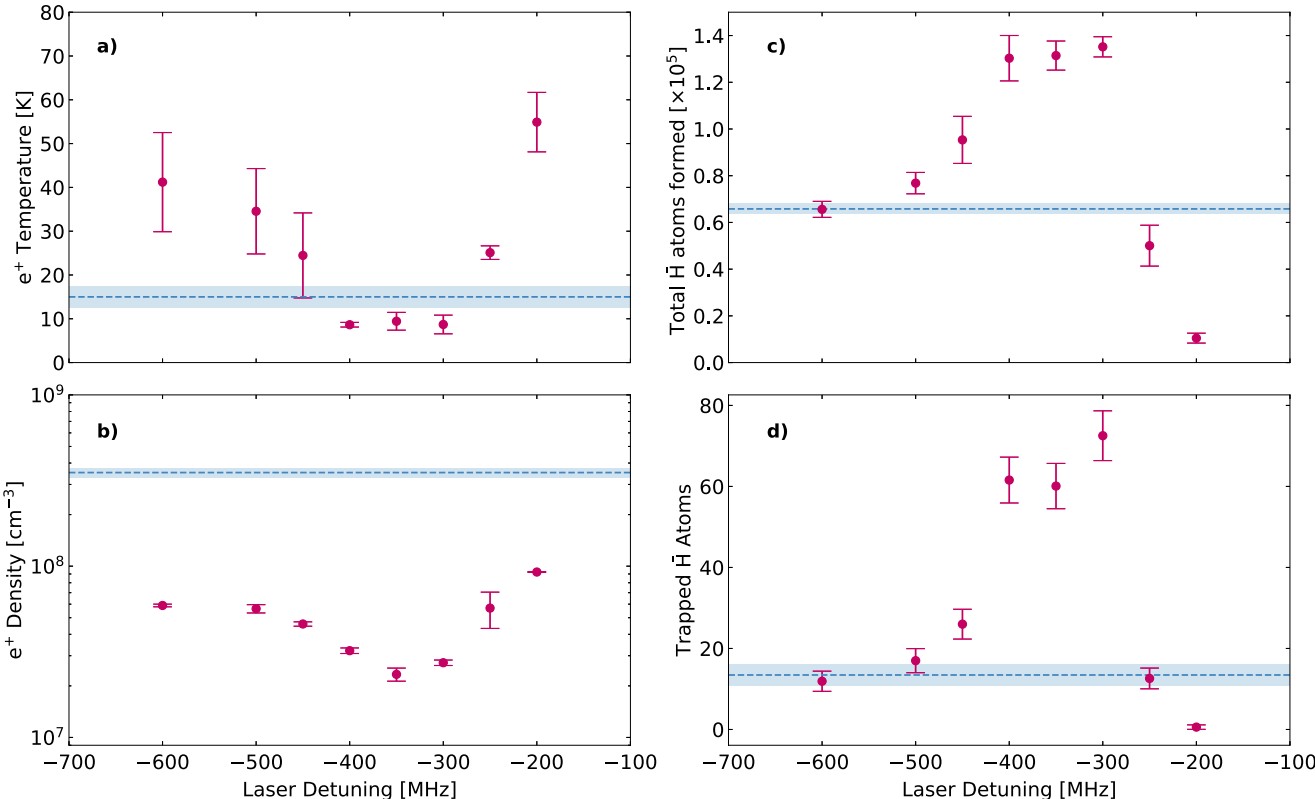

**Fig. 2 | The effect of varying the detuning of the Be⁺ cooling laser from resonance during the final step.** Measurements (purple data points) of (**a**) e⁺ plasma temperature, (**b**) e⁺ plasma density, (**c**) number of H̄ atoms synthesised during mixing (units of 10⁵ atoms), and (**d**) number of H̄ atoms trapped per trial. The reference data for H̄ synthesis without using Be⁺ (blue dashed lines, with one standard deviation indicated by the light blue band) are those that were measured initially during the same experimental period, but do not represent the slightly

higher performance reported in Ahmadi et al.[8]. Here, the final laser cooling was done with a laser power of ~160 mW along the off-axis path (cf. Fig. 1). About $1.3 \times 10^5$ Be⁺ ions were used to sympathetically cool $3.4 \times 10^6$ e⁺. The e⁺ temperature and radius were measured at the step immediately before the Be⁺/e⁺ were merged with p̄. The radii of the e⁺ plasmas corresponding to the densities measured are 0.3–0.5 mm (numbers of e⁺ were constant). The error bars indicate one standard deviation of the mean of multiple measurements for each detuning.

temperature strongly influences both the synthesis efficiency and the temperature of the resulting H̄ because the p̄ tend to thermalise with the e⁺ before they combine[7,13,18]. A lower H̄ temperature means that a larger fraction is trappable, while the synthesis efficiency will saturate at 100%, which is consistent with the peak we see in Fig. 2c (we observe a negligible number of p̄ left in the trap after synthesis at the lowest e⁺ temperatures). Note that the lowest e⁺ temperatures measured here are near the threshold of the technique. The ratio of trapped to synthesised H̄ is still small, though improved by a factor of ~3 as the detuning changes the efficacy of Be⁺ cooling. It is important to note that the e⁺ density also changes with the e⁺ temperature, and it is possible that some of the change in trapping efficiency stems from this effect. However, in previous work[7], where we have operated at lower e⁺ densities (but without Be⁺), we did not observe such strong effects from similar e⁺ density changes. Further studies should eventually allow us to disentangle the effect of density changes from the effect of temperature changes. Combining a range of studies, we have plotted the fraction of synthesised H̄ trapped versus the measured e⁺ temperature in Fig. 3, where we compare the measurements to a simulation.

Despite the variation in e⁺ density over the e⁺ temperature range investigated, we observe a clear and consistent trend that decreased e⁺ temperature increases the trapped fraction. For comparison, we also show a simulation of trapping using an improved calculation of the type discussed by Jonsell et al.[13] (Methods). The primary improvement is an increase in the number of p̄ trajectories simulated, which allows for sufficient statistics to be able to analyse only H̄ with kinetic energy <0.5 K, thus making the rescaling procedure in Jonsell et al.[13]

unnecessary. The simulation result shown here does include an additional scaling factor of 1.4 to account for the spin-orbit interaction, which causes 70% of the nascent anti-atoms to be diamagnetic, trappable, low-field seekers, as discussed by Robicheaux[33]. The simulation and our results are of the same order of magnitude and show a similar overall trend, though the simulation predicts a stronger dependence on temperature than observed and a lower trapped fraction in the range studied. Simplifications in the simulations likely play a role here; for example, the simulation does not include the detailed potential manipulations during synthesis that result in an expansion of the e⁺ plasma (cf. ref. 7). Additionally, cooling that arises from the radiative cascade of H̄ from its initial excited state down to the trapped ground state[34,35] is not included in the simulation. Hence, the H̄ atoms are assumed to retain their kinetic energy in their final trapped state, a feature that likely contributes to the lower trapped fraction in the simulations. An additional complication is the aforementioned, unquantified systematic shifts of our temperature diagnostic that could influence the comparison. Further investigations of this are beyond the scope of this article, but the overall agreement supports the expectation from the simulations that, in our current parameter regime, the p̄ thermalise with e⁺ before forming H̄.

Following the successful application of our new procedure for H̄ synthesis and trapping, we combined it with our accumulation procedure, in which we leave the trapping magnetic fields energised while repeatedly synthesising H̄, thereby accumulating H̄ from multiple synthesis cycles[7]. A single synthesis procedure uses one ejection of p̄ from ELENA but requires two ELENA cycles (each ~120 s) to complete, due to the time required for preparing our plasmas. Thus, new H̄ atoms

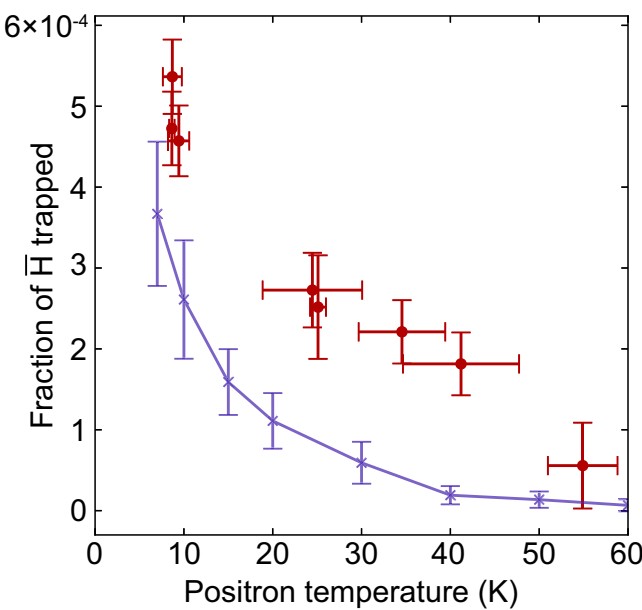

**Fig. 3 | Antihydrogen trapping efficiency vs e⁺ temperature.** The red points are experimental data whose error bars represent one standard deviation of multiple measurements. The trapped fraction is defined as the number of H̄ trapped per H̄ formed. For the experimental data, the SVD triggers during synthesis are used as a proxy for H̄ formed. The final laser cooling was off-axis, and the detuning was set between −600 MHz and −200 MHz to achieve a variation of the e⁺ temperature. The laser power was 160 mW. As a result, the e⁺ densities varied between $2 \times 10^7$ cm⁻³ and $1 \times 10^8$ cm⁻³ for this dataset. The simulated results (purple connected points) were calculated using the method described by Jonsell et al.[13] (see text), where the errors are counting errors in the simulation. The simulated e⁺ density was varied with e⁺ temperature, following the experimental results in Fig. 2. In the simulation, only H̄ surviving a 2 V/cm axial electric field were counted in order to estimate the field ionisation losses caused by the confining potentials used for the charged particles. The simulation result was subsequently scaled by 1.4 to reflect the slight prevalence of trappable states (low-field seekers) to untrappable states expected from simulations of the radiative decay processes following synthesis[33].

are added to the trap about every four minutes. Figure 4 illustrates the progression of our accumulation procedure as we developed and improved the technique. Note that due to technical and intrinsic limitations of the plasma temperature diagnostic, no e⁺ temperatures are available for this dataset.

Figure 4 shows results from approximately three months of experimentation in 2023, during which the Be⁺ procedure for synthesising H̄ was improved as losses induced by the synthesis preparations were systematically eliminated (red, blue, and green curves). The figure also shows the performance after an additional month of tuning in 2024 (orange curve). The accumulation (H̄ added per synthesis cycle) and loss rates have been quantified by fitting to the function,

$$N(x) = \frac{\Gamma}{\eta}(1 - e^{-\eta x}) \qquad (1)$$

where $N$ is the number of H̄ accumulated, $x$ is the number of synthesis cycles, $\Gamma$ is the number of H̄ added per synthesis cycle, and $\eta$ is the fractional loss per synthesis cycle. For comparison, the red curve in Fig. 4 shows the result from our standard accumulation in 2023, in which, as in previous work, e⁺ cooling relies entirely on a combination of cyclotron cooling, adiabatic expansion cooling, and evaporative cooling. The blue and green curves (also from 2023) use similar Be⁺/e⁺ mixtures and similar laser detuning for the final on-axis laser cooling (−600 MHz and −500 MHz, respectively).

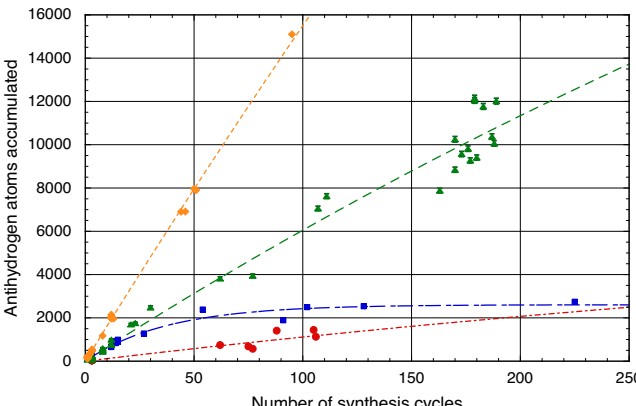

**Fig. 4 | Evolution of the H̄ accumulation rate, 2023 and 2024.** The red curve (circles) shows the non-Be⁺ assisted accumulation first prepared in 2023. The blue curve (squares) shows H̄ accumulation using the initially developed Be⁺ assisted procedure (July 2023). The green curve (triangles) shows the final Be⁺ assisted procedure after several loss-inducing steps had been removed in 2023. The orange curve (diamonds) shows accumulation in 2024 after additional improvements (see text) and a doubling of the p̄ number used for synthesis. Each dataset has been fit with a function that describes a constant rate of H̄ trapping per synthesis cycle and a fixed fractional loss of trapped H̄ per cycle (see text). The average time between individual cycles is four minutes. For the red, blue, green, and yellow curves, the fitted accumulation rates were $12 \pm 1$, $69 \pm 1$, $65 \pm 1$, and $163 \pm 1$ H̄ per cycle, respectively, and corresponding losses were $0.15 \pm 0.12\%$, $2.7 \pm 0.1\%$, $0.13 \pm 0.01\%$, and $0.10 \pm 0.03\%$ of the trapped H̄ per cycle.

Trapped H̄ can annihilate on residual gas, but it can also be lost due to interactions with the various charged particles (e⁻, e⁺, Be⁺, and p̄) we manipulate as part of the synthesis and accumulation process (*cf.* discussion in Ref. 18). The saturation of accumulation evidenced by the blue curve in Fig. 4 is dominated by H̄ losses caused by our synthesis procedure. We identified several procedural steps that led to losses that could be eliminated (*e.g.*, moving millions of Be⁺ ions across the trap (Methods)). The green curve shows data taken after the removal of the main sources of H̄ loss. In this final procedure, the time spent by charged particles in the magnetic trap region was minimised, apart from the final laser cooling of Be⁺/e⁺ (3 s duration) and the actual synthesis step, where the p̄ are merged with the Be⁺/e⁺ mixture (2–3 s duration). Large variations remained and were due to instability of our laser detuning and cycle-to-cycle timing. These two issues were addressed in early 2024 (orange curve), when we also switched to using the off-axis laser (detuning −300 MHz) and doubled the number of p̄ entering the synthesis (the e⁺ density was $1.3 \times 10^8$ cm⁻³ at the start of synthesis for these experiments). The final loss rates (green and orange curves in Fig. 4) are indistinguishable from those observed without Be⁺ and consistent with a separately estimated H̄ lifetime due to annihilations on residual gas of around 50 h (which equals ~0.1% loss per synthesis cycle). This lifetime changes due to variations in the vacuum conditions during the months when the data in Fig. 4 were taken.

## Discussion

Having removed most identified limiting factors, we maintained a near constant accumulation rate of $163 \pm 1$ H̄ atoms per four-minute cycle for up to seven hours, resulting in a record of more than 15000 H̄ atoms simultaneously trapped. (Note that not every cycle succeeds− due to intermittent problems with the accelerator complex or our apparatus). This is about eight times our best reported rate and number using non-Be⁺ assisted synthesis. This spectacular result is due to the combination of using sympathetically cooled e⁺ and an increase in p̄ available for synthesis. The direct translation of a doubling of p̄

number into a doubling of trapped $\bar{\text{H}}$ was unexpected, as previous attempts with non-Be$^+$ assisted synthesis only resulted in a scaling factor of about 1.3. We speculate that the active sympathetic cooling during the synthesis process is more efficient at maintaining a low temperature during the synthesis process than the previous method (EVC) and is thus the key to the observed absolute increase in numbers.

To put these numbers in perspective, our 2018 characterisation of the 1S-2S transition in $\bar{\text{H}}$ used about 16000 trapped $\bar{\text{H}}$ atoms from 1032 synthesis cycles and took 10 weeks to accomplish[8]. Combined with a change in measurement protocol[36] and the Be$^+$ assisted synthesis described here, a measurement of equal or better statistical power can now be accomplished in less than a day. In fact, 15000 trapped atoms is equivalent to about half of the total number of trapped antihydrogen atoms used for all published measurements to date. The procedure has allowed us to accumulate $\bar{\text{H}}$ for experiments on a near daily basis during $\bar{\text{p}}$ beam time, facilitating a range of measurements using in total more than 2 million trapped $\bar{\text{H}}$ over the 2023 and 2024 experimental runs (results from all of these experiments will be analysed and published separately). Note that we have regularly applied laser cooling[9] to the accumulated sample, typically cooling for a few hours before undertaking a given experiment. In addition, following the initial success in 2023, we expanded the setup during 2024 to apply Be$^+$ assisted synthesis in our gravity apparatus, ALPHA-g[10], and observed an initial improvement in the accumulation rate of more than a factor of 20. The current work thus facilitates large systematic studies of important parameters. It also opens the door to study sidereal and other variations that require fast (relative to a year) accumulation of data. It further enables various and new aspects of $\bar{\text{H}}$ physics, such as the dynamics of laser cooling and adiabatic cooling[37] or transitions between excited states, to be studied at a faster pace. In a larger perspective, the increased production rate itself should aid efforts towards producing beams and eventually fountains of $\bar{\text{H}}$[38,39]. Our recent measurement of the influence of gravity on $\bar{\text{H}}$[10] will also greatly benefit from the improvements in trapping rate demonstrated here. Finally, further increases in the number of $\bar{\text{p}}$ available for synthesis are expected, as our current apparatus has not yet been fully optimised to utilise the full potential of the ELENA $\bar{\text{p}}$ beam. The potential for Be$^+$ assisted $\bar{\text{H}}$ synthesis to drive further breakthroughs is clear.

## Methods

### Elimination of $\bar{\text{H}}$ losses during accumulation

Following the first implementation of Be$^+$ assisted $\bar{\text{H}}$ accumulation, we found that while the accumulation rate was higher, the amount of accumulated $\bar{\text{H}}$ eventually saturated (blue curve in Fig. 4). After a careful review of the annihilations observed by our silicon vertex detector (SVD) during accumulation, we identified several sources of $\bar{\text{H}}$ losses that could be eliminated by changing procedures. Figure 5 shows four examples of such losses, three of which were eliminated during this study.

The possibility of direct $\bar{\text{H}}$ loss from collisions of $\bar{\text{H}}$ with Be$^+$ ions was already identified by Madsen et al. [18]. However, at the time, accumulation of $\bar{\text{H}}$ was not a consideration, so the mechanism was considered negligible. When accumulating large quantities of $\bar{\text{H}}$ as we did here, the presence of charged particles in the $\bar{\text{H}}$ trap volume can cause significant losses though collisions that cause annihilations or heating of the $\bar{\text{H}}$. The losses described in Fig. 5a–c were all eliminated by changing the various procedures used to reduce the presence of e$^-$, e$^+$, or Be$^+$ in the $\bar{\text{H}}$ trap. The losses in Fig. 5d remain but are negligible compared to the losses from collisions with the residual gas in our system that limit the lifetime of $\bar{\text{H}}$ to about 50 h. A 50 h lifetime corresponds to a fractional loss of 0.13% per synthesis cycle – consistent with our best accumulation performance (Fig. 4).

### Experimental setup for Be$^+$ ablation and laser cooling

The ALPHA-2 apparatus[40] is housed in an experimental zone in CERN's Antiproton Decelerator facility and, due to radiation hazards, is closed off for access when receiving $\bar{\text{p}}$. The pulsed 355 nm laser for producing Be$^+$ and the continuous wave 313 nm laser for laser cooling Be$^+$ are housed in a temperature-controlled room (Laser Laboratory) adjacent to the experimental zone (Fig. 6). Since it can take many hours to accumulate the large number of $\bar{\text{H}}$ atoms necessary for precision measurements, the system for Be$^+$ ablation and laser cooling is entirely remotely controlled. In particular, the 313 nm light is transmitted in free space along actively stabilised optical paths to maintain beam pointing stability over long periods of $\bar{\text{H}}$ accumulation.

The 355 nm light is generated as the third harmonic of a pulsed Nd:YAG Ultra20 laser built by Quantel. The laser produces 6.3 ns pulses with 2.3 mJ of energy. A half-wave plate in combination with a polarising beam splitter (PBS) is used to split the 355 nm pulse so that most of the pulse energy is sent to a beam dump and the remainder of the pulse is directed to the experimental zone (Fig. 6). The half-wave plate is mounted in a motorised rotation stage, which enables remote adjustment of the ratio of the energies of the pulses emerging from the PBS. In the experimental zone, the pulse is split again by a 50:50 beam splitter (BS) so that half the pulse is sent to a pyroelectric sensor to monitor pulse energy and the other half is sent through a $f = 25$ cm plano-convex lens, which focuses the beam on to a beryllium metal target in the vacuum chamber. Pulse energies incident on the beryllium target were typically 40–60 μJ, corresponding to fluences of ~2–3 J cm$^{-2}$.

A TOPTICA TA-FHG Pro system is used to produce 313 nm light from two frequency doubling cavities applied to a 1252 nm amplified laser diode. A small fraction of the 626 nm light from the first doubling cavity in the TA-FHG Pro is fibre coupled to a HighFinesse WSU-2 wavelength meter for monitoring the laser frequency. The 313 nm frequency is adjusted and stabilised with a PID feedback loop applied to the grating piezo of the 1252 nm fundamental laser diode. Similar to the 355 nm system, a half-wave plate in combination with a PBS is used to split the 313 nm beam into the on-axis path and the off-axis path (cf. Figs. 1, 6). The half-wave plate is mounted in a motorised rotation stage, which allows for dynamic adjustment of the ratio of the powers in the on-axis and off-axis beams during experimentation. Remotely controlled shutters are used to control 313 nm exposure to the apparatus. The on-axis and off-axis beams are each stabilised with a combination of two piezo-actuated mirrors and two position-sensitive detectors (PSD) that operate with PID feedback loops. In each beam path, the first PSD is a low-power detector designed to monitor the power and 2D position of the ~1% of light that is transmitted through one of the mirrors in the beam path. The second PSD is a high-power thermopile detector that monitors the power and 2D position of the beam exiting the apparatus and also acts as a beam terminator. Typical powers measured by the second PSD in each beam path were 130 mW in the on-axis beam and 160 mW in the off-axis beam. The beam waists in the centre of the trap were ~2 mm. The 313 nm light is circularly polarised by quarter-wave plates before the 313 nm beams enter the vacuum chamber.

### SDR-EVC: Stabilisation of the number of Be$^+$ ions used for sympathetic cooling

Laser ablation used to produce Be$^+$ plasmas results in fluctuations in the number of trapped Be$^+$ ions of up to a factor of 10. Control of the plasma parameters, such as number and density, is critical for obtaining reproducible Be$^+$ plasmas and subsequently for achieving reproducible, cold Be$^+$/e$^+$ plasmas for $\bar{\text{H}}$ synthesis. In previous work[22], the Strong-Drive Regime–EVaporative Cooling (SDR-EVC) technique was developed to enable reproducible control of e$^+$ and e$^-$ plasma parameters, as briefly discussed below.

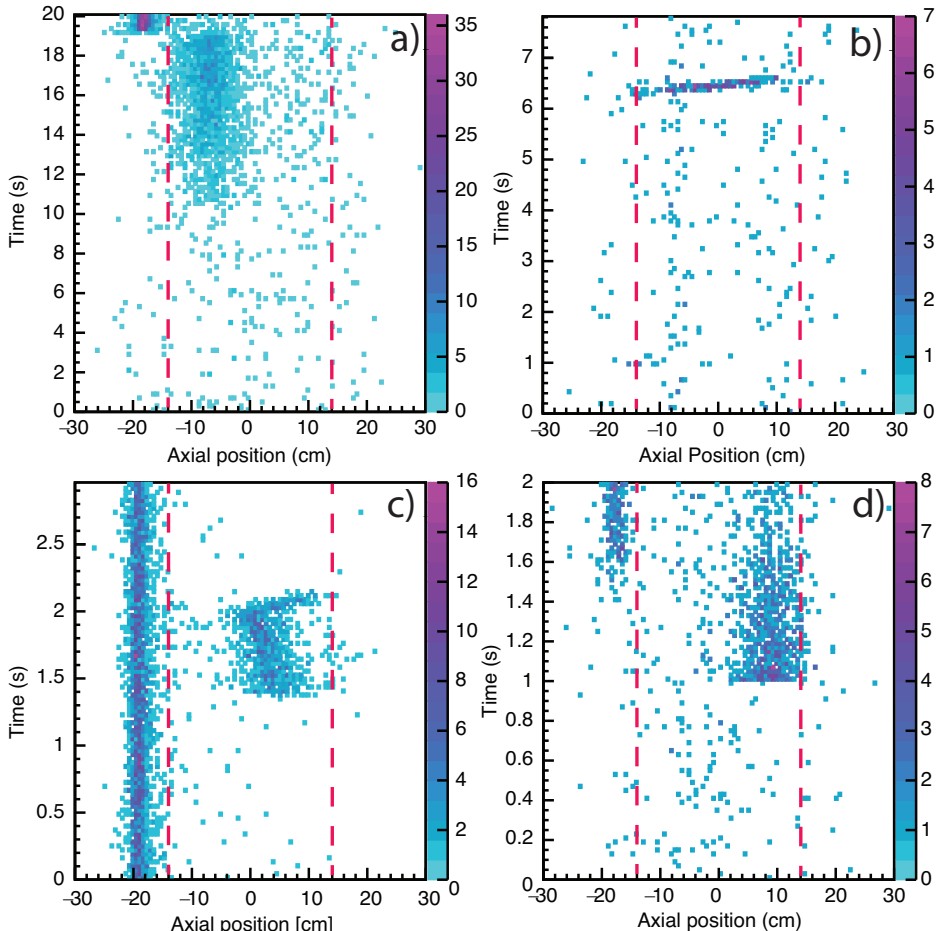

**Fig. 5 | Antiproton annihilations observed by the SVD during various phases of the H̄ accumulation process.** Each plot looks at a specific time window during a synthesis cycle and shows the integration of detected annihilations in that time window over many synthesis cycles. Therefore, the total 'time' of each plot varies with the number of synthesis cycles that have been integrated. The red dashed lines indicate the axial extent of the magnetic-minimum trap where H̄ is accumulated. **a** Annihilations during a single experiment where 2835 H̄ were accumulated in 225 synthesis cycles. The time window includes SDR-EVC on e⁻ and the transfer of p̄ into the left trap region (Fig. 1). The peak at −19 cm (also visible in (**c**) and (**d**)) is p̄ annihilating on residual gas during their preparation in the left trap region. The smeared-out peak around −7 cm from 10 s to 19 s are H̄ annihilating due to e⁻ that were ejected to the right (i.e., into the H̄ trap region) during SDR-EVC on e⁻. This H̄ loss was removed by letting the SDR-EVC process spill e⁻ to the left (negative

positions), away from the trapped H̄. **b** Annihilations during two separate experiments that together accumulated 9715 H̄ in 199 synthesis cycles. The 8 s window covers a ~ 0.5 s move of ~5 million Be⁺ from the left trap to the right trap, and we see a trail of H̄ annihilations during this movement. These losses were removed by moving the initial Be⁺ preparation from the right trap to the left trap such that only ~100k Be⁺ ions are ever present in the magnetic-minimum trap region. **c** Annihilations during one experiment that accumulated 9274 H̄ in 159 synthesis cycles. The losses occur while merging Be⁺ with e⁺. They were eliminated by merging them in the right trap outside the magnetic-minimum trap. **d** Annihilations during eight separate experiments that accumulated a sum total of 92779 H̄ in 1443 synthesis cycles. The H̄ losses (region 4–16 cm, 1–2 s) are due to the presence of recaptured e⁺ that are held briefly (for ~1 s) in the magnetic-minimum trap region until they cool into a potential well in the right trap.

In the zero-temperature limit, the plasma rotation frequency, $f$, and on-axis self-potential, $\phi_c$, of an infinitely long cylindrical plasma column are given by,

$$f = \frac{e}{4\pi\varepsilon_0 B} n \qquad (2)$$

and,

$$\phi_c = \frac{n e r_p^2}{4\varepsilon_0}\left[1 + 2\ln\left(\frac{R_W}{r_p}\right)\right] \qquad (3)$$

where $\varepsilon_0$ is the permittivity of free space, $n$ is the density of the plasma, $e$ is the elementary charge, and $B$ is the strength of the magnetic field; $r_p$ and $R_W$ are the plasma and electrode radii, respectively.

The plasma rotation frequency can be controlled by applying a rotating electric field (called a rotating wall[41] to the plasma using

azimuthally segmented electrodes (orange in Fig. 1). When the rotating wall is applied in the strong-drive regime[23], the radius of the plasma compresses or expands until the rotation frequency of the plasma is nearly equivalent to the applied rotating wall frequency. Consequently, the rotating wall can be used to set the density of the plasma when operated in the strong-drive regime. The on-axis self-potential of the plasma is controlled by evaporative cooling[15], where the axial electrostatic potential barriers are lowered so that the most thermally energetic particles escape from the electrostatic confinement well.

The equation for the self-potential admits only one solution for the plasma radius when the potential barriers are reduced so that the plasma is held in a shallow electrostatic well of depth $\phi_c$ and the density is set by the rotating wall frequency. Therefore, by applying a rotating wall and changing the depth of the electrostatic confinement well, we can independently control the density and radius of the plasma, thereby also setting the number of particles in the plasma, $N$,

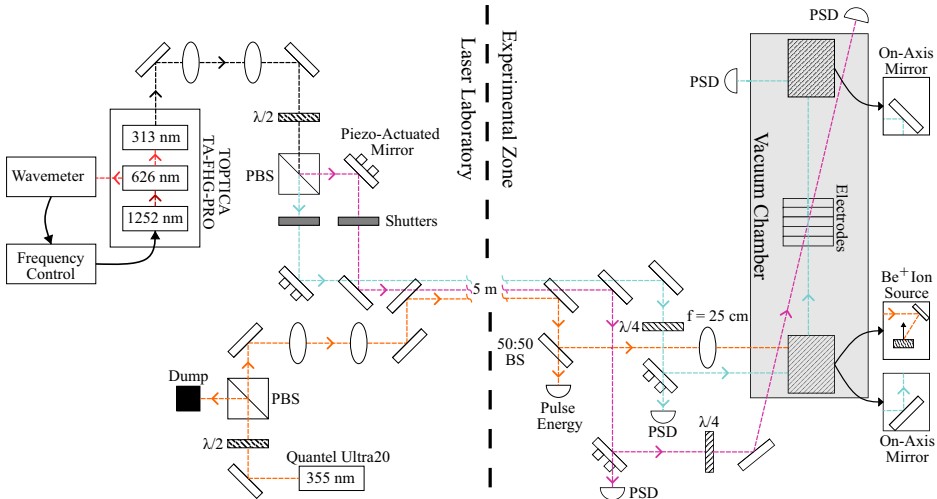

**Fig. 6 | Schematic of the optical system used for producing and laser cooling Be+.** A TOPTICA TA-FHG PRO produces 313 nm light using two frequency doubling stages with a 1252 nm amplified diode laser. A half-wave plate (λ/2) and a polarising beam splitter (PBS) split the 313 nm beam into the off-axis path (purple) and the on-axis path (cyan). Each beam path is actively stabilised with a feedback system utilising position-sensitive detectors (PSD) and piezo-actuated mirrors. Quarter-wave plates (λ/4) placed before the beams enter the vacuum chamber control polarisation. Be+ ions are produced by focusing pulsed 355 nm light from a Quantel Ultra20 laser onto a beryllium target (beam path in orange). The pulse energy is controlled with a half-wave plate and a PBS. A 50:50 beam splitter (BS) is used to direct half of the pulse to a pyroelectric sensor to monitor pulse energy. The rectangles with diagonal lines in the vacuum chamber represent linear translators, which are used to change the instruments that are aligned with the axis of the trap electrodes.

which is given by,

$$N = \int_0^V n dV = n\pi r_p^2 L \tag{4}$$

where $V$ is the plasma volume and $L$ is the length of the plasma. The plasmas in this study satisfy the condition $r_P \ll L$, so axial end effects can be neglected. We have named this technique SDR-EVC[22].

The SDR-EVC model, as described briefly above and presented in Ahmadi *et al.*[22], agrees quantitatively with the observed SDR-EVC behaviour of $e^-$ and $e^+$ plasmas. However, the radii and densities measured after the simultaneous application of the rotating wall and evaporative cooling to Be+ plasmas do not agree with those expected from assuming that the plasma rotation frequency matches the applied rotating wall frequency. Possible explanations for the discrepancy are that the rotating wall is coupling to particle motion other than the rotation of the plasma column or that the on-axis cooling laser applies additional torque to the plasma. A detailed discussion of the application of the SDR-EVC technique to Be+ plasmas is beyond the scope of this paper. Despite this disagreement, we can use the frequency of the rotating wall to control the radial size of the Be+ plasma column, and we can use the depth of the electrostatic well to control the number of particles in the Be+ plasma, in accordance with the SDR-EVC technique.

The SDR-EVC technique requires sufficient cooling to counteract heating from the rotating wall. Cooling of $e^+$ and $e^-$ plasmas is readily achieved in a strong magnetic field (1–3 T) due to emission of cyclotron radiation. For Be+ plasmas, cyclotron cooling is negligible, and we therefore use laser cooling. After ablation, the plasma temperature is ~$10^4$ K. We laser cool the plasma for 10 s at a detuning of −4 GHz, which consistently brings the plasma temperature down to ~800 K. Subsequently, we apply a rotating wall with a frequency of 150 kHz and reduce the electrostatic well depth from 15.6 V to 1.0 V in 10 s. The cooling laser remains on during this process to counteract the heating caused by the application of the rotating wall. This yields reproducible Be+ ion plasmas with 128k particles and 0.16 mm radii.

Figure 7 shows the stability of this technique over 15 days of experiments in 2023, when Be+ assisted H̄ synthesis was in use.

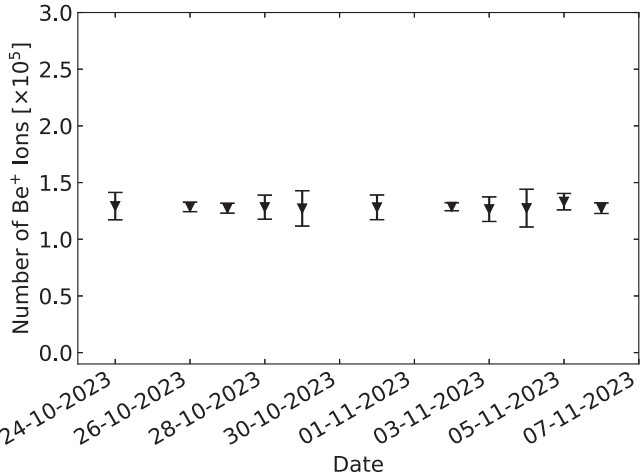

**Fig. 7 | The number of Be+ ions used for sympathetically cooling e+ over a period of 15 days, during which the Be+ assisted H̄ accumulation technique was used.** The error bars represent one standard deviation of multiple measurements performed on each day.

### Machine-learning suppression of the cosmic ray background

Antiproton annihilations are observed by our SVD, which detects the on-average 2.6 charged pions resulting from p̄ annihilating on the trap electrodes. The SVD consists of three layers of double-sided microstrip sensors, with a total of 36,864 readout channels. By reconstructing the trajectories of the pions, the positions of the annihilations (vertices) were determined with a resolution of several millimeters[27]. Data acquisition in ALPHA is implemented using MIDAS[42] (Maximum Integrated Data Acquisition System). Positron annihilations are not observed, so the SVD signal alone does not distinguish between p̄ and H̄. For the trapped H̄ results presented in this paper, we only consider annihilations during periods when no p̄ are present in the system, as has been described in previous papers (e.g., refs. 7,8,10). The main background thus consists of false positive signals from cosmic rays.

To suppress the rate of background events in the detector, a machine-learning classifier was trained to filter $\bar{\text{H}}$ events from cosmic rays, similar to previous ALPHA work. Fourteen high-level variables sensitive to topological differences between annihilation and background events were chosen as inputs to a boosted decision tree classifier implemented using ROOT's TMVA package[43]. The signal sample (613k events) was obtained from $\bar{\text{H}}$ produced during the $\bar{\text{H}}$ synthesis periods throughout 2023, filtered to only include periods of high event rates to minimise cosmic contamination. The background sample (1.14 million events) was collected when there were no $\bar{p}$ in the apparatus. A classifier cut was chosen to maximise the Punzi figure of merit[44] to five sigma over a period of 18 s, corresponding to the typical length of time it takes for the trap to release all $\bar{\text{H}}$ by ramping down the magnetic-minimum trap fields. The resulting classifier reduces the default background rate of $5.705 \pm 0.003\,\text{s}^{-1}$ to $0.0265 \pm 0.0002\,\text{s}^{-1}$ while yielding a detection efficiency of $0.721 \pm 0.001$ annihilations per readout trigger, which has been shown via simulation to be 99.5% efficient to annihilations.

### Simulations of the trapping efficiency

The simulations were carried out using the methods described by Jonsell et al.[13], where further details can be found. The three-body recombination process,

$$e^+ + e^+ + \bar{p} \rightarrow \bar{\text{H}} + e^+ \qquad (5)$$

was simulated using classical equations of motion. Since the initial states formed through the three-body process are generally very highly excited (binding energy of the order of the $e^+$ temperature), they are well described by classical physics. Trajectories of single $\bar{p}$ embedded in a $e^+$ plasma were calculated. Thus, any effects from $\bar{p}$-$\bar{p}$ interactions were neglected. The $e^+$ plasma was taken as an infinitely long cylinder with radius and density matching experimental conditions (depending on $e^+$ temperature, 0.8–1.4 mm and $1.7 \times 10^7$-$7 \times 10^7\,\text{cm}^{-3}$, respectively).

Antihydrogen atoms formed in the $e^+$ plasma eventually escape in the radial direction. After this point, the anti-atoms are followed for another 1 μs while subject to an axial electric field gradually increasing to 2 V/cm. In this way, we ensure that the atom is stable against field ionisation. The further evolution of the anti-atom is not included in the simulation; thus, we have not simulated the radiative cascade to the ground state. Instead, we select the anti-atoms in a state favourable to trapping, imposing the conditions that the canonical angular momentum is negative (i.e., the anti-atom is in a low-field seeking state) and that the kinetic energy is less than 0.5 K (the approximate trap depth). The fraction of anti-atoms was first divided by 2, assuming half of the atoms are formed in a low-field seeking spin state, and then scaled by 1.4 to account for the spin-orbit interaction, as discussed by Robicheaux[33].

To obtain sufficient statistics, between 162,500 and 650,000 $\bar{p}$ trajectories have been simulated for each $e^+$ temperature. The $\bar{p}$ were initialised with 10 meV kinetic energy in the axial direction and subsequently slowed due to interaction with the $e^+$ plasma. The axial energy distribution of the emerging $\bar{\text{H}}$ atoms was found to be near thermal with the temperature set by the $e^+$.

It should be noted that the simulations do not capture the details of the field configurations of the experiment. Other potentially important effects that are not included are the heating and expansion of the $e^+$ plasma observed during mixing with $\bar{p}$. Perhaps most importantly, the radiative cascade has not been simulated. Simulations have shown that the $\bar{\text{H}}$ atoms lose kinetic energy during this cascade[34,35], which could potentially explain why the simulated trapping fractions are lower than those observed. An additional complication is the aforementioned, unquantified systematic shifts of our temperature diagnostic that could influence the comparison. Further investigations of this are beyond the scope of this paper, but the overall agreement supports the expectation from the simulations that, in our current parameter regime, the $\bar{p}$ thermalise with $e^+$ before forming $\bar{\text{H}}$.

## Data availability

The datasets generated during and/or analysed during the current study are available from JSH (jeffrey.hangst@cern.ch) on request. Due to the inherent complexity of the experiment, it is not straightforward to analyse the multiple data sets without assistance from the collaboration. Such assistance can be provided on request.

## Code availability

Codes used for data analysis in the current work are available from J.S.H. (jeffrey.hangst@cern.ch) on request.

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

## Acknowledgements

This work was supported by: CNPq, FAPERJ, RENAFAE (Brazil); NSERC, NRC/TRIUMF, EHPDS/EHDRS, CFI, DRAC (Canada); FNU (Nice Centre), Carlsberg Foundation (Denmark); STFC, EPSRC, the Royal Society and the Leverhulme Trust (UK); DOE, NSF (USA); ISF (Israel); and VR (Sweden). The computations were enabled by resources provided by the National Academic Infrastructure for Supercomputing in Sweden (NAISS), partially funded by the Swedish Research Council through grant agreement no. 2022-06725. We thank J. J. Bollinger (NIST) for advice regarding laser cooling beryllium and useful discussions regarding nonneutral plasma dynamics in Penning-Malmberg traps. We thank the following individuals who contributed to this work as undergraduate or Master's students: R. Wilkins, H. Strojecka, E. A. Sweeney, S. M. Determan, S. E. Price.

## Author contributions

The authors are members of the ALPHA Collaboration at CERN. This experiment was based on data collected using the ALPHA-2 antihydrogen trapping apparatus. The ALPHA-2 apparatus was designed and constructed by the ALPHA Collaboration and operated with methods developed by the entire collaboration. The beryllium experiment was first suggested by N.M. The beryllium setup was designed and implemented by MBGG, S.A.J., N.M., J.P., T.R.B. and K.A.T. The beryllium experimental protocols and analysis were conceived and executed by N.M.B., M.B.G.G., N.M., T.R.B. and K.A.T. The manuscript was written by N.M. with assistance from M.B.G.G., L.M.G., J.S.H., S.J., C.Ø.R., S.J. and K.A.T. The manuscript was then edited and improved by all authors. In addition to authors identified above, the following authors participated in scientific discussions of the procedures and results and contributed to the data-taking by working shifts on the ALPHA experiment: R.A., L.O.A.A., C.J.B., W.B., G.B., A.C., I.C., C.L.C., M.C., A.C.M., A.D.V., D.D.Q., S.J.E., A.E., J.E., J.F., T.F., M.C.F., M.E.H., D.H., C.A.I., A.J.U.J., V.R.M., J.T.K.M., T.M., J.N., A.N.O., A.P., F.R., R.L.S., E.S., J.S., D.M.S., J.Singh, G.S., C.S., S.S., J.Suh, A.S., T.D.T., R.I.T., E.T.W., M.U., D.P.W., P.W. and J.S.W.

## Competing interests

The authors declare no competing interests.

## Additional information

[1]Department of Physics and Astronomy, University of British Columbia, Vancouver, BC, Canada. [2]Instituto de Fisica, Universidade Federal do Rio de Janeiro, Rio de Janeiro, Brazil. [3]Department of Physics, Faculty of Science and Engineering, Swansea University, Swansea, UK. [4]School of Physics and Astronomy, University of Manchester, Manchester, UK. [5]Cockcroft Institute, Sci-Tech Daresbury, Warrington, UK. [6]University of Brescia, Brescia and INFN Pavia, Pavia, Italy. [7]TRIUMF, Vancouver, BC, Canada. [8]Department of Physics, University of California at Berkeley, Berkeley, CA, USA. [9]Department of Physics and Astronomy, University of Calgary, Calgary, AB, Canada. [10]Department of Physics and Astronomy, Aarhus University, Aarhus, Denmark. [11]Department of Physics, Simon Fraser University, Burnaby, BC, Canada. [12]Van Swinderen Institute for Particle Physics and Gravity, University of Groningen, Groningen, The Netherlands. [13]Department of Physics, Stockholm University, Stockholm, Sweden. [14]Department of Chemistry, University of British Columbia, Vancouver, BC, Canada. [15]Experimental Physics Department, CERN, Geneva, Switzerland. [16]Department of Physics and Astronomy, Purdue University, West Lafayette, IN, USA. [17]Ben Gurion University of the Negev, Be'er Sheva, Israel. [18]Soreq NRC, Yavne, Israel. [19]INFN Pisa, Pisa, Italy. [20]Physics Department, Marquette University, Milwaukee, WI, USA. ✉e-mail: jeffrey.hangst@cern.ch; niels.madsen@cern.ch

