## [Transparent Peer Review file · Nature Communications]

Be⁺ assisted, simultaneous confinement of more than 15000 antihydrogen atoms

Corresponding Author: Professor Jeffrey Hangst

Version 0:

Reviewer comments:

Reviewer #1

(Remarks to the Author)

The manuscript submitted by the ALPHA collaboration reports on new developments to prepare a large quantity of trapped antihydrogen atoms at the AD/ELENA facility for antimatter studies. The method is based on the use of laser cooled Be⁺ ions to further cool the positron plasma which is merged with the antiprotons to synthesize antihydrogen atoms. The synthesis rate now reached 160 antihydrogen atoms / 4-minute cycle, which is 8 times better than what was used lately. A total of 15000 antihydrogen atoms trapped simultaneously after 7 hours of preparation was achieved.

The achievements are world class and make possible new investigations on synthesis and cooling processes, as well as on antihydrogen properties.

The article is straightforward, in the best meaning: it is well focused on the details of the method (steps of the cycle, cooling, number of particles in the plasma and plasma dimensions, technical information on cycle duration and lasers used,...), while the achievements are clear. The manuscript provides to the reader the necessary information to understand the relevant factors that lead to the reported achievements. The manuscript is very well written and could be published as such.

I congratulate the ALPHA collaboration for this successful development, and support the publication of these results in Nature Communication.

Still, one aspect of the submission requires an additional iteration: the figures of the Extended Material are missing in the material made accessible to the reviewer. I am therefore not able to review the Extended Material at this stage. Can the missing figures (Extended Data Fig. 1, 2, 3) be made available?

Also, I suggest the authors to consider the following two minor remarks.

The manuscript and results are of excellent quality which speak for themselves. The last sentence of the abstract contains few qualifications (“revolutionary technique fundamentally transforms”, “sidereal effects”, “plethora of fundamental studies”) which may not have the positive impact expected by the authors.

The first sentence of the manuscript focuses on “exotic atoms, or atomic systems that do not naturally occur”. The introduction indicates that the authors restrict the discussion to atomic system containing at least one antiparticle, while muonic atoms or atoms with a radioactive nucleus can also be considered as atomic systems that do not naturally exist. This first sentence may be improved if narrowed to what follows.

Reviewer #2

(Remarks to the Author)

The authors report on the sympathetic cooling of positrons and, at one remove, antiprotons, using laser-cooled beryllium ions in the ALPHA antihydrogen apparatus. The effect of the detuning of the laser used to cool the beryllium on the positron temperature, density, and the number of antihydrogen atoms formed and trapped is reported. Crucially, even though the temperature of the positron plasma is only reduced by a relatively modest factor of ~2.5, and the density is slightly reduced, the effect of the positron cooling and additional improvements in the antihydrogen state preparation is dramatic – the number produced per cycle increased by a factor of 8. Remarkably, the number of antihydrogen atoms used to perform the first 1S-

2S measurement over many months, 16000, can now be achieved in less than a day. This confirms the collaboration's insight that positron temperatures are the current bottleneck to producing larger quantities of trappable antihydrogen.

I think that the dramatic increase in antihydrogen production using laser-cooled beryllium will be of considerable significance to the field of trapped antimatter research. While there have been preliminary studies in the ALPHA apparatus of using laser-cooled Be⁺ to sympathetically cool positrons (<https://doi.org/10.1038/s41467-021-26086-1>), this is the first time that this method has been used to improve the production of trappable antihydrogen, and incorporating this method into the tremendously complex ALPHA apparatus is an impressive technical feat.

The evidence for the improvement in antihydrogen production is compelling, and the methodology is sound. ALPHA are the world leaders in the production of antihydrogen and its use in precision tests, and this is another impressive result. The paper is well written and generally very easy to follow.

I have a number of minor points which I think the authors could address to improve the manuscript.

1) On page 4: Why is the Be⁺ plasma initially so hot (500K)? Is this optimised to ensure mixing of the beryllium and positron plasmas? Or does further cooling not help for some reason? Perhaps the authors could comment on this

2) On page 4, the parallel energy analyser technique is mentioned. As this method underpins all the temperature determinations in the paper and seems to have some large potential systematic errors, I thought the description of this method was too brief. I think it would help the non-specialist reader to at least describe in a few sentences what this method is, and the authors might consider providing more details in the supplementary methods section.

3) On page 5: I think it could be even more clearly spelt out that there is an up to 50% uncertainty on the absolute temperature determination but less uncertainty on the relative temperature difference. I think it is also important for the authors to comment on the systematic errors of the method when measuring relative temperatures >5K. If they feel that there are no systematic errors, this should be justified.

4) The reduction of temperature by a factor of 2.5 is reported, but the absolute numbers are not discussed. Even considering the 50% systematic uncertainty, I think it would be clearer to report these numbers and mention the uncertainty associated with them. Also, in Figure 2 a), the black dashed line with the blue uncertainties doesn't seem, by eye, to correspond to either 18 K or 2.5 times the minimum positron temperature in Fig. 2 a), I think this could be clarified.

5) In other work, the ALPHA collaboration have demonstrated laser cooling of antihydrogen. I understood that this cooling was also continuously applied during the stacking process. I wondered if these sympathetic cooling techniques were compatible with the antihydrogen laser cooling, perhaps the authors could comment on this in the paper.

6) I think that much of the extended discussion of the elimination of antihydrogen losses during accumulation in Extended Data Fig. 1's caption should be moved to main text of the Methods section; I appreciate that enough information to understand the figure needs to be provided in the caption, but as a reader I found that having essentially no discussion of these methods in the main text made this section hard to follow.

Reviewer #3

(Remarks to the Author)

NCOMMS-25-23723-T: Manuscript review

"Be⁺ assisted, simultaneous confinement of more than 15000 antihydrogen atoms" by R. Akbari et al. (ALPHA collaboration)

Summary: The manuscript presents recent results of increasing trapped antihydrogen production rates by a factor of 8 using a sympathetic laser cooling technique that cools the positrons with laser-cooled Be⁺ ions before the synthesis of antihydrogen. The measurements are conducted in the ALPHA-2 apparatus in the Antimatter Factory of CERN by the ALPHA collaboration.

Overall, the presented manuscript has a high quality of data, is technically sound, and the data was appropriately analysed, interpreted carefully, and is well presented. The conclusions are sound and straightforwardly supported by the experimental data, and the manuscript uses appropriate references.

The work presented here represents a significant milestone for antihydrogen experiments, as limitations in the production rates and measurement statistics that limited possibilities to explore antihydrogen in precision measurements in the past decade have now finally been overcome. I congratulate the authors on this achievement. The results reported here will play a key role in all future precision measurements on antihydrogen and contribute to improved sensitivity on the discovery of potential differences between matter and antimatter. These are two strong arguments that support the publication of this manuscript in Nature Communications.

Previous work of the authors regarding this direction of research was published in Nature Communications in 2021 (Baker et al., Nat. Commun. 12, Article number: 6139 (2021), reference 8 in the presented manuscript), which reported the reduction of the positron plasma temperature using laser-cooled Be⁺ ions, but the impact on the production of antihydrogen is to my knowledge not previously reported, and represents the key result of the presented manuscript. This novel content is of great interest to a broader community, which is a further argument to support the publication of this manuscript.

I have a few questions to the presented content that I would like the authors to address and to consider for revision and improvement of their manuscript:

Section "Antihydrogen synthesis and trapping"

"...about 10⁵ antiprotons in an ellipsoidal cloud of radius ~0.4 mm with a temperature of ~100 K ..." It would be helpful here to include a reference to the temperature measurement technique used to determine the plasma temperature. Later in the manuscript, refs. [28] and [29] are given, and presumably also used here. I suggest adding a reference here.

Section "Sympathetically cooled positrons":

The authors comment on improvements of their procedure in the second and third paragraphs of this section compared to their previous work (Ref. [8]) before introducing the actual sympathetic cooling scheme in the fourth paragraph. I would like to ask the authors to reconsider the order of the presented text.

Second paragraph: The timing of the cooling/mixing cycle is not clear after reading the manuscript. The authors comment here on the improvement of the cooling cycle from 4 minutes to 100 s, but later in the manuscript, it says that the charged particles are kept away from the antihydrogen trap, and only introduced for a few seconds for the mixing procedure.

Therefore, it is not clear to what extent the laser cooling power was limiting the cycle time, and which part of the procedure was accelerated by the improvements discussed here. I suggest to present a timing protocol of the experiment sequence in the methods section to make this more clear.

A second question is how the axial laser beam reduces the cycle time? Is it related to reducing the radial component of the k-vector in the trap, or does this allow a different protocol to be executed?

Fourth paragraph: The laser cooling of the Be⁺ ions is applied with different parameters, but it is not motivated why the detuning and power are changed during the procedure. Is it of advantage to not fully compress and cool the Be⁺ plasma during the preparation procedure? Or is this to account for the initial high temperature of Be⁺ ions after the laser ablation? It is also not consequently stated which laser parameters are used, e.g. in the next section "Beryllium assisted antihydrogen synthesis", the text says "the laser-cooling is maintained" and "The final laser cooling step", but it is not clear if and how the laser parameters are adjusted.

Section: "Beryllium assisted antihydrogen synthesis"

Second paragraph:

Please add information about the laser power and laser detuning used in the procedure.

Please also comment if there is any difference in the performance using the on-axis or off-axis beam other than enabling additional diagnostics.

Third paragraph:

"In our previous work, ..." I think here the authors want to say that they used the optimized parameters reported from their previous work, but it is not explicitly stated in this way. Please rephrase this sentence to avoid ambiguity.

Section: "Accumulation of antihydrogen"

The authors comment on the vacuum conditions and the annihilation rate per cycle in their trap. I would like to ask the authors if they observe annihilation related to the laser ablation process, as this releases atoms and ions, and thus locally produces residual gas following the ablation pulse.

Another question to the procedure is how frequently are the Be⁺ ions loaded?

Are they kept/recycled or ejected after the mixing cycle?

Section: "Conclusions"

Is the apparatus operating now at a saturation value of the antiproton number, or can a further increase in trapped Hbars be expected when increasing the antiproton number further? Is the limitation known, or to be explored?

This is also a major result, that the trapped Hbar production can now be scaled to a higher number of antiprotons.

Other comments:

Higher production rates and lower antihydrogen temperatures are an interesting field for future improvements. While this may be speculative, are there any directions to be explored that the authors consider to be promising to achieve further improvements?

For example, the beryllium plasma temperature is still 3-4 orders of magnitude above the Doppler limit. Is there a plan to improve the temperature diagnostics of the beryllium ions to be able to explore sympathetic cooling at lower temperatures? Will a further reduction in the beryllium temperature be a benefit, or is the process already limited by the spatial separation of the positions and the Be⁺ ions?

Is a further increase of the antihydrogen production necessary to accelerate the antihydrogen measurement program, or do the authors consider their system to be sufficiently developed to handle the measurements outlined in the perspectives section?

If the antihydrogen production was increased by another factor of 10, which part of the experimental program becomes accessible which is currently not?

Version 1:

Reviewer comments:

Reviewer #2

(Remarks to the Author)

I am happy that my points have been substantially addressed.

1)

Original Question: On page 4: Why is the Be⁺ plasma initially so hot (500K)? Is this optimised to ensure mixing of the beryllium and positron plasmas? Or does further cooling not help for some reason? Perhaps the authors could comment on this

Authors' Reply: The Be⁺ is hot at this stage of the preparation as the rotating electric fields (rotating wall) applied during the SDR-EVC technique (Methods) heat the plasma significantly. This is a well-documented effect due to "slippage" between the plasma and the external field when in the strong drive regime. While this is of course interesting, we have not changed the text as details of the SDR-EVC technique are beyond the scope here.

Follow up comment: OK, my question was more aimed at understanding why further cooling of the Be⁺ plasma is not beneficial at this stage, but it is a minor point, and I do not think a change is necessary.

2)

Original Question: On page 4, the parallel energy analyser technique is mentioned. As this method underpins all the temperature determinations in the paper and seems to have some large potential systematic errors, I thought the description of this method was too brief. I think it would help the non-specialist reader to at least describe in a few sentences what this method is, and the authors might consider providing more details in the supplementary methods section.

Authors' Reply: The technique, that we did not develop, is very well described in the referenced papers. However, it's important to understand that for most of the antihydrogen data (essentially all of Figure 4) we have no ability to measure temperatures with said technique – so in practice it serves more as a guide to check if we're doing something sensible – the ultimate test for now is antihydrogen trapping. A fluorescence technique is being pursued in order to be able to measure relevant temperatures at the moments in the cycle when they are useful. We have added a comment in the discussion of Figure 2 to make it more explicit that these temperature measurements shouldn't be overinterpreted. We have also added a comment that we have no temperature data for the data in figure 4.

Follow up comment: OK, I think with the additional caveats now in the text this is clearer.

3)

Original Question: On page 5: I think it could be even more clearly spelt out that there is an up to 50% uncertainty on the absolute temperature determination but less uncertainty on the relative temperature difference. I think it is also important for the authors to comment on the systematic errors of the method when measuring relative temperatures >5K. If they feel that there are no systematic errors, this should be justified.

Authors' Reply: The systematic error of 50% is only in the absolute, as stated on the top of page 5. We have indirect evidence from our first sympathetic cooling demonstration that the temperature measurement method, for our parameter regime, is in good agreement with an alternative method using the plasma density distribution. In any case, we clearly state that quantitative comparison with simulation/theory is made more difficult by this problem. More studies with new temperature diagnostic methods are needed to validate the simulations we have available, something beyond the scope of this work.

Follow up comment: As with point 2), I think with the additional caveats now in the text this is clearer.

4)

Original Question: The reduction of temperature by a factor of 2.5 is reported, but the absolute numbers are not discussed. Even considering the 50% systematic uncertainty, I think it would be clearer to report these numbers and mention the uncertainty associated with them. Also, in Figure 2 a), the black dashed line with the blue uncertainties doesn't seem, by eye, to correspond to either 18 K or 2.5 times the minimum positron temperature in Fig. 2 a), I think this could be clarified.

Authors' Reply: The factor 2.5 reported is documented in detail in our previous published work, which is clearly referenced. In the current work, we don't focus on this number particularly, as the focus is on the antihydrogen synthesis – and the dependency of the trapped fraction with temperature. As described in the text, we are not able to measure temperatures lower than about 5 K with our current technique. For the experiments where we do have all the information possible, we show the details in Fig. 2. In order to avoid confusion about this point we've removed the explicit reference to factor 2.5 from the abstract.

Follow up comment: This has removed my confusion, thank you.

5)

Original Question: In other work, the ALPHA collaboration have demonstrated laser cooling of antihydrogen. I understood that this cooling was also continuously applied during the stacking process. I wondered if these sympathetic cooling techniques were compatible with the antihydrogen laser cooling, perhaps the authors could comment on this in the paper.

Authors' Reply: We have added a comment in the conclusion. The short answer is that while we have demonstrated laser-cooling during accumulation, it's not technically reliable when accumulating for many hours. Cold samples can be prepared in a few hours following the accumulation – which is much more sparing on the hardware and staff.

Follow up comment: Thank you, this make sense, I can see that the methods are not in conflict in this case.

6)

Original Question: I think that much of the extended discussion of the elimination of antihydrogen losses during accumulation in Extended Data Fig. 1's caption should be moved to main text of the Methods section; I appreciate that enough information to understand the figure needs to be provided in the caption, but as a reader I found that having essentially no discussion of these methods in the main text made this section hard to follow.

Authors' Reply: As is clear from Figure 4, reducing losses of already trapped antihydrogen induced by the synthesis preparations and process are key to good accumulation. As is mentioned, we identified several sources of loss, some of which were mostly bad strategies/mistakes (such as spilling millions of electrons into the antihydrogen trap region) and our key message here (and with the methods) is that having large amounts of antihydrogen around allowed us to seek these out and eliminate them. A detailed discussion of each and every problem we found doesn't seem to be merited, as long as awareness and a rough idea of the type of problems is addressed as it is now.

Follow up comment: My concern as a reader was not so much with the overall level of detail, which I agree should not include a detailed discussion of each and every problem, but rather that I found that too much of the information appeared in the caption and not enough was in the main text. I leave it up to the Editors if they want to address this, as it is ultimately a decision about presentation rather than science.

Reviewer #3

(Remarks to the Author)

I would like to thank the authors for the discussion of the questions provided in the response letter, and for providing further inside into the procedures and perspectives of their work.

I am looking forward on future reports on further refinement of the presented method.

The manuscript has been improved to a level where the publication of these important results should no longer be delayed. I recommend the editors to make a positive decision without further revisions of the manuscript.

Response to referee comments
(in bold)

Reviewer #1 (Remarks to the Author):

The manuscript submitted by the ALPHA collaboration reports on new developments to prepare a large quantity of trapped antihydrogen atoms at the AD/ELENA facility for antimatter studies. The method is based on the use of laser cooled Be⁺ ions to further cool the positron plasma which is merged with the antiprotons to synthesize antihydrogen atoms. The synthesis rate now reached 160 antihydrogen atoms / 4-minute cycle, which is 8 times better than what was used lately. A total of 15000 antihydrogen atoms trapped simultaneously after 7 hours of preparation was achieved.

The achievements are world class and make possible new investigations on synthesis and cooling processes, as well as on antihydrogen properties.

The article is straightforward, in the best meaning: it is well focused on the details of the method (steps of the cycle, cooling, number of particles in the plasma and plasma dimensions, technical information on cycle duration and lasers used,...), while the achievements are clear. The manuscript provides to the reader the necessary information to understand the relevant factors that lead to the reported achievements. The manuscript is very well written and could be published as such.

I congratulate the ALPHA collaboration for this successful development, and support the publication of these results in Nature Communication.

Still, one aspect of the submission requires an additional iteration: the figures of the Extended Material are missing in the material made accessible to the reviewer. I am therefore not able to review the Extended Material at this stage. Can the missing figures (Extended Data Fig. 1, 2, 3) be made available?

We thank the reviewer for the positive evaluation of our work. It is up to the Editor to supply the figures, as we have no direct contact with the referees.

Also, I suggest the authors to consider the following two minor remarks.

The manuscript and results are of excellent quality which speak for themselves. The last sentence of the abstract contains few qualifications (“revolutionary technique fundamentally transforms”, “sidereal effects”, “plethora of fundamental studies”) which may not have the positive impact expected by the authors.

We believe that the technique described here is truly a new paradigm for our field. In our opinion, this is the single biggest development since the demonstration of antihydrogen trapping itself in 2010.

The first sentence of the manuscript focuses on “exotic atoms, or atomic systems that do not

naturally occur". The introduction indicates that the authors restrict the discussion to atomic system containing at least one antiparticle, while muonic atoms or atoms with a radioactive nucleus can also be considered as atomic systems that do not naturally exist. This first sentence may be improved if narrowed to what follows.

We have added a reference to muonic atoms.

Reviewer #2 (Remarks to the Author):

The authors report on the sympathetic cooling of positrons and, at one remove, antiprotons, using laser-cooled beryllium ions in the ALPHA antihydrogen apparatus. The effect of the detuning of the laser used to cool the beryllium on the positron temperature, density, and the number of antihydrogen atoms formed and trapped is reported. Crucially, even though the temperature of the positron plasma is only reduced by a relatively modest factor of ~ 2.5 , and the density is slightly reduced, the effect of the positron cooling and additional improvements in the antihydrogen state preparation is dramatic – the number produced per cycle increased by a factor of 8. Remarkably, the number of antihydrogen atoms used to perform the first 1S-2S measurement over many months, 16000, can now be achieved in less than a day. This confirms the collaboration's insight that positron temperatures are the current bottleneck to producing larger quantities of trappable antihydrogen.

I think that the dramatic increase in antihydrogen production using laser-cooled beryllium will be of considerable significance to the field of trapped antimatter research. While there have been preliminary studies in the ALPHA apparatus of using laser-cooled Be⁺ to sympathetically cool positrons (<https://doi.org/10.1038/s41467-021-26086-1>), this is the first time that this method has been used to improve the production of trappable antihydrogen, and incorporating this method into the tremendously complex ALPHA apparatus is an impressive technical feat.

The evidence for the improvement in antihydrogen production is compelling, and the methodology is sound. ALPHA are the world leaders in the production of antihydrogen and its use in precision tests, and this is another impressive result. The paper is well written and generally very easy to follow.

We thank the reviewer for the careful review and their kind remarks on our work.

I have a number of minor points which I think the authors could address to improve the manuscript.

1) On page 4: Why is the Be⁺ plasma initially so hot (500K)? Is this optimised to ensure mixing of the beryllium and positron plasmas? Or does further cooling not help for some reason? Perhaps the authors could comment on this

The Be⁺ is hot at this stage of the preparation as the rotating electric fields (rotating wall) applied during the SDR-EVC technique (Methods) heat the plasma significantly. This is a well-documented effect due to “slippage” between the plasma and the external field when in the strong drive regime. While this is of course interesting, we have not changed the text as details of the SDR-EVC technique are beyond the scope here.

2) On page 4, the parallel energy analyser technique is mentioned. As this method underpins all the temperature determinations in the paper and seems to have some large potential systematic errors, I thought the description of this method was too brief. I think it would help the non-specialist reader to at least describe in a few sentences what this method is, and the authors might consider providing more details in the supplementary methods section.

The technique, that we did not develop, is very well described in the referenced papers. However, it’s important to understand that for most of the antihydrogen data (essentially all of Figure 4) we have no ability to measure temperatures with said technique – so in practice it serves more as a guide to check if we’re doing something sensible – the ultimate test for now is antihydrogen trapping. A fluorescence technique is being pursued in order to be able to measure relevant temperatures at the moments in the cycle when they are useful. We have added a comment in the discussion of Figure 2 to make it more explicit that these temperature measurements shouldn’t be overinterpreted. We have also added a comment that we have no temperature data for the data in figure 4.

3) On page 5: I think it could be even more clearly spelt out that there is an up to 50% uncertainty on the absolute temperature determination but less uncertainty on the relative temperature difference. I think it is also important for the authors to comment on the systematic errors of the method when measuring relative temperatures >5K. If they feel that there are no systematic errors, this should be justified.

The systematic error of 50% is only in the absolute, as stated on the top of page 5. We have indirect evidence from our first sympathetic cooling demonstration that the temperature measurement method, for our parameter regime, is in good agreement with an alternative method using the plasma density distribution. In any case, we clearly state that quantitative comparison with simulation/theory is made more difficult by this problem. More studies with new temperature diagnostic methods are needed to validate the simulations we have available, something beyond the scope of this work.

4) The reduction of temperature by a factor of 2.5 is reported, but the absolute numbers are not discussed. Even considering the 50% systematic uncertainty, I think it would be clearer to report these numbers and mention the uncertainty associated with them. Also, in Figure 2 a), the black dashed line with the blue uncertainties doesn’t seem, by eye, to correspond to either 18 K or 2.5 times the minimum positron temperature in Fig. 2 a), I think this could be clarified.

The factor 2.5 reported is documented in detail in our previous published work, which is clearly referenced. In the current work, we don't focus on this number particularly, as the focus is on the antihydrogen synthesis – and the dependency of the trapped fraction with temperature. As described in the text, we are not able to measure temperatures lower than about 5 K with our current technique. For the experiments where we do have all the information possible, we show the details in Fig. 2. In order to avoid confusion about this point we've removed the explicit reference to factor 2.5 from the abstract.

5) In other work, the ALPHA collaboration have demonstrated laser cooling of antihydrogen. I understood that this cooling was also continuously applied during the stacking process. I wondered if these sympathetic cooling techniques were compatible with the antihydrogen laser cooling, perhaps the authors could comment on this in the paper.

We have added a comment in the conclusion. The short answer is that while we have demonstrated laser-cooling during accumulation, it's not technically reliable when accumulating for many hours. Cold samples can be prepared in a few hours following the accumulation – which is much more sparing on the hardware and staff.

6) I think that much of the extended discussion of the elimination of antihydrogen losses during accumulation in Extended Data Fig. 1's caption should be moved to main text of the Methods section; I appreciate that enough information to understand the figure needs to be provided in the caption, but as a reader I found that having essentially no discussion of these methods in the main text made this section hard to follow.

As is clear from Figure 4, reducing losses of already trapped antihydrogen induced by the synthesis preparations and process are key to good accumulation. As is mentioned, we identified several sources of loss, some of which were mostly bad strategies/mistakes (such as spilling millions of electrons into the antihydrogen trap region) and our key message here (and with the methods) is that having large amounts of antihydrogen around allowed us to seek these out and eliminate them. A detailed discussion of each and every problem we found doesn't seem to be merited, as long as awareness and a rough idea of the type of problems is addressed as it is now.

Reviewer #3 (Remarks to the Author):

NCOMMS-25-23723-T: Manuscript review

“Be + assisted, simultaneous confinement of more than 15000 antihydrogen atoms” by R. Akbari et al. (ALPHA collaboration)

Summary: The manuscripts presents recent results of increasing trapped antihydrogen production rates by a factor of 8 using a sympathetic laser cooling technique that cools the positrons with laser-cooled Be⁺ ions before the synthesis of antihydrogen. The measurements are conducted in the ALPHA-2 apparatus in the Antimatter Factory of CERN by the ALPHA collaboration.

Overall, the presented manuscript has a high quality of data, is technically sound, and the data was appropriately analysed, interpreted carefully, and is well presented. The conclusions are sound and straightforwardly supported by the experimental data, and the manuscript uses appropriate references.

The work presented here represents a significant milestone for antihydrogen experiments, as limitations in the production rates and measurement statistics that limited possibilities to explore antihydrogen in precision measurements in the past decade have now finally been overcome. I congratulate the authors on this achievement. The results reported here will play a key role in all future precision measurements on antihydrogen and contribute to improved sensitivity on the discovery of potential differences between matter and antimatter. These are two strong arguments that support the publication of this manuscript in Nature Communications.

Previous work of the authors regarding this direction of research was published in Nature Communications in 2021 (Baker et al., Nat. Commun. 12, Article number: 6139 (2021), reference 8 in the presented manuscript), which reported the reduction of the positron plasma temperature using laser-cooled Be⁺ ions, but the impact on the production of antihydrogen is to my knowledge not previously reported, and represents the key result of the presented manuscript. This novel content is of great interest to a broader community, which is a further argument to support the publication of this manuscript.

We thank the referee for the positive comments.

I have a few questions to the presented content that I would like the authors to address and to consider for revision and improvement of their manuscript:

Section “Antihydrogen synthesis and trapping”

“...about 10⁵ antiprotons in an ellipsoidal cloud of radius ~0.4 mm with a temperature of ~100 K ...” It would be helpful here to include a reference to the temperature measurement technique used to determine the plasma temperature. Later in the manuscript, refs. [28] and [29] are given, and presumably also used here. I suggest adding a reference here.

We have added a reference.

Section “Sympathetically cooled positrons”:

The authors comment on improvements of their procedure in the second and third paragraphs of this section compared to their previous work (Ref. [8]) before introducing the actual sympathetic cooling scheme in the fourth paragraph. I would like to ask the authors to reconsider the order of the presented text.

Second paragraph: The timing of the cooling/mixing cycle is not clear after reading the

manuscript. The authors comment here on the improvement of the cooling cycle from 4 minutes to 100 s, but later in the manuscript, it says that the charged particles are kept away from the antihydrogen trap, and only introduced for a few seconds for the mixing procedure. Therefore, it is not clear to what extent the laser cooling power was limiting the cycle time, and which part of the procedure was accelerated by the improvements discussed here. I suggest to present a timing protocol of the experiment sequence in the methods section to make this more clear.

As explained below (and now highlighted in the text) the key upgrade was the on-axis laser that allowed shortening the preparation time, as well as the increased laser power that allowed shorter cooling periods. However, many small adjustments have been made. A timing protocol of the sequence will not add much understanding to this and is very elaborate to explain, considering that the protocol is to a large extent driven by practical issues as well as physics issues. For example, the laser-cooling on-axis is only available during periods when the translators at either end of the apparatus can be moved to the mirror position (i.e., when no particles are being transferred in or out of the apparatus). Also, all of our activities have to be well integrated with the CERN accelerator repetition rate, and our positron accumulation cycle. While these details are certainly important, they are not, we believe, of general interest and would fit better in an instrumentation paper.

A second question is how the axial laser beam reduces the cycle time? Is it related to reducing the radial component of the k -vector in the trap, or does this allow a different protocol to be executed?

The benefit of the on-axis laser is that it allows us to apply the SDR-EVC technique to stabilise the number of Be^+ ions. This is detailed in the section about the SDR-EVC technique. In our previous work, we had to do some rather complicated gymnastics to prepare the Be^+ ions in a reasonably reproducible manner, as we did not have the on-axis laser. We have added a sentence to clarify this.

Fourth paragraph: The laser cooling of the Be^+ ions is applied with different parameters, but it is not motivated why the detuning and power are changed during the procedure. Is it of advantage to not fully compress and cool the Be^+ plasma during the preparation procedure? Or is this to account for the initial high temperature of Be^+ ions after the laser ablation? It is also not consequently stated which laser parameters are used, e.g. in the next section “Beryllium assisted antihydrogen synthesis”, the text says “the laser-cooling is maintained” and “The final laser cooling step”, but it is not clear if and how the laser parameters are adjusted.

We felt that explaining all the individual tunings here would overload the paper with details, which is why we prefer the current formulation. This is the first time the SDR-EVC technique has been applied with laser-cooling, and there are still many questions about it that we’re exploring (as mentioned in the Methods). Some problems are also likely artefacts of our apparatus, and would be different when implemented in a

differen trap. To answer the questions: The ions are initially very hot after ablation (>10000K) – and not in the ground state – hence the large detuning of the cooling lasers. The SDR-EVC technique heats the plasma, and we found that too small of a detuning resulted in the heating overcoming the laser-cooling – and the SDR-EVC failing (details for another publication). As for the compression, we’re targeting a given density, but we’re combating plasma expansion when we hold plasmas, so several of the steps are influenced by the exact timing of the full procedure. We understand the confusion between “final step” and “maintained” (used in the next section) – and have changed “maintained” to be “applied at a fixed small detuning” – to mirror the description in the sympathetic cooling procedure to clarify the last question.

Section: “Beryllium assisted antihydrogen synthesis”

Second paragraph:

Please add information about the laser power and laser detuning used in the procedure. Please also comment if there is any difference in the performance using the on-axis or off-axis beam other than enabling additional diagnostics.

The power is the same throughout – and is mentioned at the beginning of the last paragraph in the previous section. We have not observed any statistically significant differences between on- and off-axis cooling during synthesis, though there’s anecdotal evidence that the off-axis laser is better (Since it’s anecdotal, and could just be a question of lack of tuning the physical overlap (position of the laser), we feel it’s an overreach to say anything either way).

Third paragraph:

“In our previous work, ...” I think here the authors want to say that they used the optimized parameters reported from there previous work, but it is not explicitly stated in this way. Please rephrase this sentence to avoid ambiguity.

Indeed – we have added a few words here to make this clear.

Section: “Accumulation of antihydrogen”

The authors comment on the vacuum conditions and the annihilation rate per cycle in their trap. I would like to ask the authors if they observe annihilation related to the laser ablation process, as this releases atoms and ions, and thus locally produces residual gas following the ablation pulse.

We have not observed such effects, but this could be due to our ablation source being positioned about 1 m away from the cold UHV region. There are also a number of apertures along the way, making the solid angle very small. We intentionally designed the ablation source (Ref [20]) to have a very small “exit hole” to reduce its residual gas footprint.

Another question to the procedure is how frequently are the Be⁺ ions loaded?
Are they kept/recycled or ejected after the mixing cycle?

A new batch is loaded every cycle to simplify the procedure and allow destructive post-synthesis diagnostics (extraction of particles to detectors) to monitor the process.

Section: "Conclusions"

Is the apparatus operating now at a saturation value of the antiproton number, or can a further increase in trapped Hbars be expected when increasing the antiproton number further? Is the limitation known, or to be explored?

We have not observed any saturation effects yet from antiproton numbers. We have a number of known things we can do to increase the antiproton numbers and they are in the works.

This is also a major result, that the trapped Hbar production can now be scaled to a higher number of antiprotons.

We absolutely agree!

Other comments:

Higher production rates and lower antihydrogen temperatures are an interesting field for future improvements. While this may be speculative, are there any directions to be explored that the authors consider to be promising to achieve further improvements?

We have only begun to explore the dependence on positron density, and there's evidence that this strongly influences the final result. The combination of the SDR-EVC technique (on both Be⁺ and e⁺) as well as the sympathetic cooling allows us to explore this, which we intend to do soon.

For example, the beryllium plasma temperature is still 3-4 orders of magnitude above the Doppler limit. Is there a plan to improve the temperature diagnostics of the beryllium ions to be able to explore sympathetic cooling at lower temperatures? Will a further reduction in the beryllium temperature be a benefit, or is the process already limited by the spatial separation of the positions and the Be⁺ ions?

The temperature question is tricky, as our diagnostic is currently at its limit. We have an upgrade on the way to install *in-situ* photon diagnostics that should allow us to measure Be⁺ temperatures by fluorescence, thus expanding the range of temperatures we can measure and aiding in the understanding of what may be required. For example, we currently do not chirp the lasers, and we only have a limited understanding of how the laser angle and position with respect to the plasma influence the final result. Simulations [see reference 19] indicate that there isn't much to gain (in fact it gets worse) by having significantly colder Be⁺ ions due to the spatial separation. However,

we speculate that introducing a mediator species (e.g. Be^{++} ions, protons, H_2^{+} ...) could improve the coupling and thereby improve the cooling.

Is a further increase of the antihydrogen production necessary to accelerate the antihydrogen measurement program, or do the authors consider their system to be sufficiently developed to handle the measurements outlined in the perspectives section?

As we explore new types of experiment we have found that each increase in production has allowed us to do more, so there's no reason to believe that the current improved production will stay "sufficient" in the longer term. We have recently been obtaining mK level antihydrogen using adiabatic expansion of the confinement well. This results in particle loss, so there is always incentive to trap more. See below also.

If the antihydrogen production was increased by another factor of 10, which part of the experimental program becomes accessible which is currently not?

This is rather speculative, but a simple example is that one way to get colder atoms than laser-cooling can deliver would be to just throw away the hot ones (recall no collisions, so this is not evaporative cooling, just velocity selection). So a factor of 10 would allow us to work with significantly colder samples of antihydrogen, and thereby improve spectroscopic and gravitational measurements accordingly.